# Does Oral Hypofunction Promote Social Withdrawal in the Older Adults? A Longitudinal Survey of Elderly Subjects in Rural Japan

**DOI:** 10.3390/ijerph17238904

**Published:** 2020-11-30

**Authors:** Yoko Hasegawa, Ayumi Sakuramoto-Sadakane, Koutatsu Nagai, Joji Tamaoka, Masayuki Oshitani, Takahiro Ono, Takashi Sawada, Ken Shinmura, Hiromitsu Kishimoto

**Affiliations:** 1Department of Dentistry and Oral Surgery, Hyogo College of Medicine, 1-1 Mukogawa-cho, Nishinomiya, Hyogo 663-8501, Japan; ayu.cherry.ayu@gmail.com (A.S.-S.); joji1122@hyo-med.ac.jp (J.T.); m.os26gjam3@gmail.com (M.O.); kisihiro@hyo-med.ac.jp (H.K.); 2Division of Comprehensive Prosthodontics, Faculty of Dentistry & Graduate School of Medical and Dental Sciences, Niigata University, Niigata 951-8514, Japan; ono@dent.niigata-u.ac.jp; 3Department of Physical Therapy, School of Rehabilitation, Hyogo University of Health Sciences, 1-3-6 Minatojima, Chuo-ku, Kobe, Hyogo 650-8530, Japan; nagai-k@huhs.ac.jp; 4Hyogo Dental Association, 5-7-18 Yamamoto-dori, Chuo-ku, Kobe, Hyogo 650-0003, Japan; sawada@fc.hda.or.jp; 5Department of General Internal Medicine, Hyogo College of Medicine, 1-1 Mukogawa-cho, Nishinomiya, Hyogo 663-8501, Japan; ke-shimmura@hyo-med.ac.jp

**Keywords:** social withdrawal, oral hypofunction, older adult, oral frailty, frailty

## Abstract

It is often assumed that oral hypofunction is associated with social withdrawal in older adults because decreased motor function is related to decreased oral function. However, few studies have examined the relationship between social withdrawal in older adults and oral function. This longitudinal study aimed to clarify the relationship between changes in the level of social withdrawal and oral function in independent older adults. Participants were 427 older adults aged 65 years or older who took part in a self-administered questionnaire from 2016 to 2017 (baseline), and again two years later (follow-up). At baseline, 17 items related to oral function and confounding factors related to withdrawal, physical condition, physical function, and cognitive function were evaluated. A Cox proportional hazard model was used to examine the oral functions that negatively impact social withdrawal. The following factors were significantly associated with the worsening of social withdrawal: the number of remaining teeth, gingival condition, occlusal force, masticatory efficiency, and items related to swallowing and dry mouth. Older adults with cognitive issues who walk slowly and have a weak knee extension muscle were also significantly more likely to have oral frailty. Those who were found to have oral frailty at baseline were 1.8 times more likely to develop withdrawal compared to those with robust oral function. The results indicated that the worsening of withdrawal was associated with oral hypofunction at baseline. Since oral hypofunction was associated with the worsening of social withdrawal in older adults, it is important to maintain older adults’ oral function.

## 1. Introduction

Falling is one of the most serious events experienced by older adults, and such events often entail expensive medical bills. Studies have found that approximately one-third of older adults who live in their own home fall at least once a year, and older adults who have fallen at least once are more prone to repeated falls. [1,2,3] Older adults who have fallen not only suffer from functional disorders but also tend to withdraw and limit daytime activities due to fear of additional falls [3]. Prolonged social withdrawal may cause sarcopenia, which is a condition characterized by muscle weakness; sarcopenia may lead to disuse syndrome, causing the patient to be bedridden [4,5]. One study found that withdrawn older adults who do not go out and constantly stay at home are significantly likely to develop a condition that requires long-term care [6]. Another study found that prolonged social withdrawal is a risk factor for long-term care and mortality [7]. To extend the healthy life expectancy of older adults, it is important to reduce their need for long-term care and prevent them from socially withdrawing.

The concept of oral frailty, which indicates a poor oral health condition, has been of interest to the medical community in recent years [8]. Oral frailty refers to a slightly impaired oral function and compromised oral hygiene, which are early symptoms of general physical frailty. General frailty progresses if oral frailty does not improve [8]. Therefore, proactive intervention to prevent oral frailty is recommended [9]. These results are consistent with the findings described in existing literature, in that frailty is associated with the number of teeth [8,10,11,12], occlusal force [12,13], masticatory performance [10,12,14], and oral hygiene [9,15].

Oral hypofunction is a dental disease in which oral function is multiply decreased due to not only aging but also various factors such as other diseases and disorders. If oral hypofunction is left untreated, masticatory dysfunction and dysphagia can develop, which can lead to malnutrition [11], which results in frailty, namely physical vulnerability [16,17]. Studies have also found that aspiration, as well as eating and swallowing disorders, are risk factors for the worsening of physical frailty [18,19], and oral hypofunction in older adults leads to disease onset and an increased mortality rate [8]. Therefore, it can be assumed that maintaining good oral function in older adults is necessary for maintaining their physical function. Mikami and colleagues [20] reported an association between oral function and the frequency of going out in older adults, suggesting that there is a relationship between social withdrawal and oral function. Withdrawal in older adults is caused by psychological, social, and physical factors that decrease their motivation and ability to go out [21]. Meanwhile, it is assumed that psychological factors accompanying impaired dental aesthetics as well as reduced mastication, swallowing, and articulation functions and halitosis from periodontal disease [22] are related to social withdrawal in older adults.

Since 2016, we have been examining the relationship between fall-related physical functions and oral functions in older adults living in the mountainous Tamba-Sasayama area, Hyogo, Japan. Our earlier studies found that participants who had experienced falls and were anxious about falling had not only reduced motor function but also impaired occlusion; such participants also had oral hypofunction, and older adults with few remaining teeth and impaired occlusion were prone to falls [12,23]. Based on these findings, we suggest that maintaining healthy oral and physical function can reduce falls and extend healthy life expectancy. Conversely, to the best of our knowledge, no studies have longitudinally analyzed the relationship between oral function and social withdrawal; there is a dearth of evidence related to this issue.

Therefore, this study examined the relationship between social withdrawal and oral function in independent older adults aged 65 years or older who live in the Tamba-Sasayama area. Specifically, we conducted a two-year follow-up survey on changes in participants’ frequency of going out and physical activity.

## 2. Materials and Methods

### 2.1. Study Participants

This prospective longitudinal survey was approved by the institutional review board of Hyogo College of Medicine (approval no. Rinhi 0342) and is part of the Frail Elderly in the Sasayama-Tamba Area (FESTA) study. From April 2016 to December 2019, a medical-dental joint academic survey was conducted with independent older adults who were living in Tamba-Sasayama City, Hyogo Prefecture. A total of 976 older adults aged 65 years or older took part in the baseline survey, of whom 428 participated (43.9% of baseline participants) in both the baseline and follow-up survey and comprised the study sample. Participants were recruited through advertisements in local newspapers and poster announcements from the Hyogo College of Medicine, Sasayama Medical Center. We voluntarily contacted only those subjects who did not apply in two years after the baseline survey. Participation in the survey was voluntary.

Exclusion criteria were as follows: those who did not give consent to undergo oral function examination and those for whom body composition analysis could not be performed as they had undergone pacemaker implantation. Participants received an explanation of the purpose and methods of the examination and provided their written consent prior to participation.

### 2.2. Survey Questionnaire

Participants completed the Kihon checklist (KCL, Appendix A) [24]. The Kihon checklist is a self-administered evaluation created by the Ministry of Health, Labor and Welfare of Japan and is a screening tool introduced when the long-term care insurance system was revised in 2006. The Kihon checklist’s questionnaire consists of 25 questions in the following 7 areas: activities of daily living, social activities of daily living, physical function, nutritional status, oral function, social activities of daily living (social withdrawal), cognitive function, and depressed mood. In the present study, we selected two social withdrawal-related questions and three questions about the oral function related to oral frailty to create our evaluation items. A score of 1 was assigned for a negative response to each of the following social withdrawal-related questions: “Do you go out at least once a week?” (Yes = 0/No = 1); and “Compared to the last year, do you go out less frequently?” (Yes = 1/No = 0) Those who obtained a total score of 1 or greater were considered to have a social withdrawal tendency.

Survey items related to oral frailty were evaluated. Answers to the following three questions were used as indicators for the masticatory function, swallowing function and dry mouth, respectively: “Compared to half a year ago, do you find it difficult to eat hard food?”; “Do you choke on tea or soup?”; and “Do you feel discomfort from dry mouth?”

### 2.3. Evaluation of Cognitive and Physical Factors

Cognitive function was evaluated using the Mini-Mental State Examination (MMSE) [25].

Skeletal muscle was evaluated through bioelectrical impedance analysis (BIA), using a body composition analyzer (InBody770, InBody Japan). The skeletal muscle mass index was calculated and used as an indicator of the skeletal muscle mass of the extremities [26]. Additionally, the body mass index was calculated based on measured body height and weight.

During the walking speed test, participants were asked to walk at a speed at which they normally walk. The normal walking speed (m/sec) (hereinafter referred to as the walking speed) was then analyzed. Since the participants might have walked faster at the beginning of the walking test and slower towards the end of the test, the walking distance was set at 7 m, which allowed for 1 m of acceleration and 1 meter of deceleration in a 10 m measurement zone [27].

A muscle strength dynamometer (Mobie, SAKAI Medical Co., Ltd., Tokyo, Japan) was used to measure knee extension muscle strength. Participants were asked to sit upright with the knees flexed at 90 degrees. It was ensured that the hips were not lifted off the treatment bench during the measurement. Measurement was performed two times for each participant with their dominant leg, and the maximum torque was evaluated. The moment arm (distance from the center of the knee joint to the measuring device) at the time of measurement was 0.25 m [28].

### 2.4. Evaluation of Oral Functioning

The oral condition and function, are components of the oral hypofunction and were evaluated by the following items: the number of remaining teeth, oral hygiene, oral moisture, occlusal force, masticatory performance, swallowing ability, speech motor control, salivary bacterial count, and answers to the questionnaire about concerning the oral functions in Kihon check list (KCL).

Items related to oral hypofunction were evaluated as described below, with reference to the criteria by the Japanese Society of Gerodontology [13].

Residual tooth root and third molar tooth were included when counting the number of remaining teeth (hereinafter referred to as the number of teeth). Those with 20 or fewer teeth were assessed as having a small number of remaining teeth. The level of oral hygiene was assessed using an oral assessment chart (Table 1) [15].

Each of the nine oral hygiene items was categorized into one of three levels: no issues = ◦, requiring caution = Δ, and with issues = ×. Participants who were assigned “×” or “Δ” were considered as having oral hypofunction. We examined dry mouth conditions by measuring twice the oral moisture of the dorsum of the tongue and the buccal mucosa using an oral moisture meter (Mucas^®^, LIFE Co., Ltd., Saitama, Japan). Those with a measurement value of <27 for both the buccal mucosa and the tongue were assessed as having oral hypofunction.

An occlusal force meter (Occlusal Force-Meter GM10, Nagano Keiki Co., Ltd.) was used in measuring the maximum occlusal force of the locations equivalent to the left and right first molars. The left and right sides were measured separately, and the sum of the values for both sides was evaluated [12,23]. For those who have lost the first molar, the measurement was performed at a location closest to that of the first molar, where the participant could hold the device between the upper and lower teeth. For those who were using dentures, the measurement was performed while they were wearing the dentures. Those with an occlusal force value of <30 kgf, which was the lowest 20% of the total participants at baseline, were considered as having oral hypofunction.

A gummy jelly was used in measuring masticatory performance. Participants were asked to spit out the test gummy jelly after chewing 30 times. Masticatory performance was evaluated on a 10-point scale (from 0 for the minimum to 9 for the maximum) [10]. Those with a score of 2 or below were considered as having oral hypofunction. In the Repetitive Saliva Swallowing Test (RSST), participants were asked to swallow saliva as many times as possible for 30 s. Those with a score of less than 3 were considered as having oral hypofunction. For tongue pressure, the maximum tongue pressure was measured two times using a JMS tongue pressure measurement device (JMS tongue pressure meter, JMS, Hiroshima, Japan). The maximum value was used for the analysis. Those with a tongue pressure value of 30 kPa or below were considered as having oral hypofunction. For oral diadochokinesis, participants were asked to repeatedly pronounce the syllable, “ta” in five seconds. An automatic measurement device (Kenko–Kun Handy, Takei Scientific Instruments Co., Ltd.) was used in counting how many times the participant could pronounce the syllable. Those who pronounced the syllable less than six times per second were considered as having oral hypofunction. For oral bacterial counts, a bacterial counter (Panasonic Healthcare Co., Ltd., Tokyo, Japan) was used to evaluate the bacterial count on the tongue surface. Those with a bacterial level of 4 or above were considered as having oral hypofunction [15].

Participants sat in a reclining nursing chair. Oral condition was examined in artificial light at sufficient intensity.

### 2.5. Statistical Analysis

The results of the baseline examination were examined and compared to the follow-up examination two years later. Participants whose social withdrawal condition was found to have worsened at the follow-up compared to baseline were allocated to the “worse” group, and all other participants were allocated to the “unchanged/improved” group.

The relationship between the oral function evaluation items and the level of social withdrawal were tested at baseline using a chi-square test respectively. Items with *p* < 0.1 were selected as contributing factors of oral frailty. Additionally, the relationship between social withdrawal-related physical factors and oral frailty was tested using the Kruskal–Wallis test and multiple comparisons.

Subsequently, a Cox proportional hazard model was used to test the relationship between social withdrawal and oral hypofunction using forced entry into the model. Duration (the number of days) from baseline to the second examination was set as the time variable, the status of social withdrawal as the objective variable, and the state of oral frailty as well as factors with a significant association with oral frailty as the explanatory variables. The level of significance was set at *p* < 0.05 for all variables.

## 3. Results

Data at baseline are shown in Table 2.

One participant did not answer the questionnaire on social withdrawal at baseline, responses from the remaining 427 participants were therefore analyzed at baseline. Items examined for oral function that were found to be associated with social withdrawal were as follows: the number of remaining teeth, gingival condition, occlusal force, masticatory efficiency, and items related to swallowing and dry mouth in the KCL. In other words, participants who were assessed as withdrawn had oral hypofunction.

The relationship between social withdrawal-related cognitive/physical factors and oral frailty at baseline is shown in Table 3.

There was a significantly higher tendency for older adults with impaired cognitive function (lower MMSE score) who walk slowly and have weak knee extension muscle strength to have oral frailty. Since these physical factors could be confounding factors strongly associated with social withdrawal, they were used as explanatory variables in the Cox proportional hazards model.

Table 4 shows the results of the change of social withdrawal situation.

Two participants did not respond to the second survey, therefore three were excluded from the longitudinal analysis, including one who did not respond at baseline. As a result, a total of 425 subjects were able to evaluate changes in social withdrawal.

At follow-up, 12.9% (*n* = 55) of the participants newly exhibited a social withdrawal tendency, while 67.2% (*n* = 286) did not do so. Moreover, 8% (*n* = 34) of the participants were assessed as withdrawn at baseline and the social withdrawal did not improve at follow-up.

The relationship between social withdrawal and oral frailty was examined using a Cox proportional hazard model (Table 5).

The results of the longitudinal examination indicated that those with oral frailty at baseline were 1.8 times more likely to develop social withdrawal compared to those with robust oral function. In addition, a slow walking speed was found to be a significant risk factor for social withdrawal.

## 4. Discussion

The results of this study indicated that the worsening of social withdrawal was associated with oral hypofunction (oral frailty) at baseline. In addition, it was shown that the subjects with oral frailty often had an impaired physical ability and low cognitive function and that a poor physical function was involved in the worsening of social withdrawal.

### 4.1. Oral Frailty, Oral Hypofunction, and Social Withdrawal

The results of this study indicated that the following factors that lead to a reduced eating and swallowing function were significant risk factors for oral frailty: a small number of remaining teeth; reduced occlusal force and masticatory performance; difficulty in swallowing and dry mouth; and poor oral hygiene. These results are consistent with existing literature in that frailty is associated with the number of teeth [8,10,11,12], occlusal force [12,13], masticatory performance [10,12,14], and oral hygiene [9,15]. In this study, oral frailty was confirmed based on the methods by Tanaka et al. [8]. In clinical dentistry in Japan, a patient with progressed oral frailty is diagnosed with oral hypofunction, which requires therapeutic intervention [13]. A patient is diagnosed with oral hypofunction when he satisfies three out of the following seven assessment items: oral uncleanness; oral dryness; decline in occlusal force; decline in motor function of tongue and lips; decline in tongue pressure; decline in chewing function; and decline in swallowing function. These items were included in the items for oral function associated with social withdrawal in this study. In other words, it was suggested that poor oral function at baseline may be associated with the worsening of a social withdrawal tendency.

### 4.2. Relationship between Oral Hypofunction and Social Withdrawal

This study also indicated that not only age and physical abilities, but also impaired cognitive function was a factor associated with oral frailty. Tooth loss is associated with poorer cognitive function in elderly population studies [29,30,31]. While it has been found that causes of impaired cognitive function are numerous and include: aging; diseases such as hypertension, hyperlipidaemia, and hyperglycaemia; stress, insomnia; chronic pain; and depression [32,33], oral frailty may also be associated with impaired cognitive function. Additionally, it has been suggested that socializing and community participation are key to the prevention of social withdrawal. One study reported that in order to prevent dementia, it is important to continue to socialize and engage in physical activities [34]. Conversely, opportunities for going out often involve eating and drinking. Those with eating and swallowing disorders can eat limited types of food and may need to pause during meals. Such people are likely to experience a psychological burden as a result of going out, which prevents them from having a meal with others and leads to reduced communication skills and potential worsening of social withdrawal.

In the present study, subjects with social withdrawal at baseline had lower physical abilities than those without, and our results indicated that risk factors for the worsening of social withdrawal were slow walking speed and oral frailty. Poor oral health in older adults is known to be associated with disability and declining physical function [35]. The finding that oral frailty is associated with the worsening of social withdrawal suggests that providing dental intervention in addition to existing interventions for preventing cognitive impairment [34] may be effective in preventing the worsening of social withdrawal in older adults. To date, studies have reported that: a decline in masticatory function in older adults can be prevented through regular dentist visits [36]; dental treatment can improve the nutritional condition and increase vegetable and fruit intake [37]; and the improvement of oral frailty is associated with the prevention of the progression of the frailty cycle [17]. Hence, to prevent the worsening of social withdrawal, it is important to maintain oral function.

### 4.3. Limitations

One of the limitations of this study was that it evaluated social withdrawal based on responses given by participants to the questionnaire. In other words, the evaluation was based on the subjective perceptions of participants. It is possible to evaluate the actual degree of social withdrawal by using an objective indicator such as an activity monitor. Thus, there is room for improvement in the methods of evaluating social withdrawal.

### 4.4. Clinical Implications

Sherrington and colleagues [38] have reported that an intervention involving a physical exercise program is effective in reducing falls in older adults and helps facilitate the improvement of their physical and psychosocial health. However, it is essential for older adults to eat nutritious food, improve muscle strength, and go out to maintain their activity levels. Research has shown that the causes of social withdrawal include not only physical factors but also psychosocial ones [6,21]. This study found that oral hypofunction is one such cause, which suggests that maintaining good oral function in older adults not only prevents disease onset and an increase in mortality [8] but also may contribute to the prevention of social withdrawal and have a positive impact on the physical and psychological state of health. The improvement of the oral function is reportedly important for improving the QOL of the elderly [39,40], so maintaining or improving the oral function is important for preventing social withdrawal and improving the QOL.

## 5. Conclusions

The results of this study showed that not only reduced physical function but also oral hypofunction is associated with the worsening of social withdrawal in older adults, suggesting the importance of maintaining their oral function.

## Figures and Tables

**Table 1 ijerph-17-08904-t001:** Clinical Oral Assessment Chart.

Clinical Oral Assessment Chart			
Item	◦: **No Problem**	Δ: **Cautious**	×: Problematic
	Participants continue current care	Caregivers consider asking a specialist for assessment when no improvement is seen	Participants need treatment or intervention by a specialist
Mouth opening	Participants easily open mouth for care	Participants refuse to open mouth Caregivers can open mouth manually with 2 fingerbreadths	Caregivers open mouth with <1 fingerbreadth because of tooth clenching and contracture of temporomandibular joint
Bad breath	None	Caregivers sense bad breath when approaching the oral cavity	Caregivers sense a smell of bad breath in a room
Drooling	None	Decline in swallowing reflex is suspected but no drooling	Drooling (because of decline in swallowing reflex)
Dryness of mouth and saliva	No friction in mucosa on palpation with gloved fingers Mucosa has saliva	Slightly increased friction, no tendency for the gloved fingers to adhere to the mucosa Mucosa has little saliva and is sticky	Significantly increased friction, gloved fingers adhering to the mucosa Mucosa has little saliva and is dry
Teeth and dentures	Clean and no plaque and debris No mobile teeth	Small amount of plaque and debris Several mobile teeth but no hindrance to care	Large amount of plaque and debris Some wobbly teeth
Oral mucosa	Pink and moist No dirtiness	Dry and color change such as reddening	Spontaneous bleeding, ulcer, and candida infection are observed Airway secretion, desquamated epithelium, and clotting blood are apparent and tightly attached to the mucosa
Tongue	Moderate filiform papillae present	Extension and loss of filiform papillae (coated tongue and bald tongue, respectively	
Lips	Smooth (no cracking)	Cracked and angular cheilitis	
Gingiva	Tightened (stippling)	Gingiva is swollen and bleeds while brushing	

The assessments were categorized into qualitative variables: ◦ for “No problems” and Δ or × for “Cautious/Problematic.” according to the Clinical Oral Assessment Chart [15].

**Table 2 ijerph-17-08904-t002:** A summary of participant information and the relationship between social withdrawal and oral function.

Baseline Survey		Not-Withdrawal	%	Withdrawal	%	*p*-Value
Participants		337	78.9	90	21.1	
*General*						
Age (mean ± S.D.)		72.2 ± 5.6		74.0 ± 6.2		0.012
Gender *	Male	130	86.1	21	13.9	0.007
	Female	207	75.0	69	25.0	
*Oral condition*						
Remaining teeth (<20)		105	31.2	37	41.1	0.079
State of oral hygiene	Mouth opening	13	3.9	3	3.3	0.814
(assigned “×” or “Δ” by Clinical Oral Assessment Chart)	Bad breath	22	6.5	5	5.6	0.733
	Drooling	0	0.0	0	0.0	-
	Dryness of mouth and saliva	69	20.5	13	14.4	0.197
	Teeth and dentures	70	20.8	19	21.1	0.944
	Oral mucosa	17	5.0	4	4.4	0.813
	Tongue	34	10.1	10	11.1	0.777
	Lips	6	1.8	1	1.1	0.642
	Gingiva *	24	7.1	13	14.4	0.029
Oral moisture (<27)	Buccal mucosa	119	35.3	35	37.8	0.665
	Dorsum of the tongue	169	50.1	52	57.8	0.198
Occlusal force (<30 kgf) *		58	17.2	25	27.8	0.024
Masticatory performance (score < 3) *		76	20.1	27	30.4	0.040
RSST (<3 times/30 sec)		35	10.4	9	10.1	0.927
Tongue pressure (<30 Kpa)		124	36.8	36	40.0	0.577
Oral diadochokinesis“ta” (6 >/sec)		55	16.3	20	22.2	0.190
Salivary bacterial count (4 < Level )		292	91.3	83	94.3	0.350
KCL assessment	Masticatory function	8	2.4	5	5.6	0.146
	Swallowing function *	20	6.0	17	18.9	*p* < 0.001
	Dry mouth *	17	5.0	20	22.2	*p* < 0.001

Data at baseline are shown. Oral condition: Measured items (cut-off values for oral hypofunction [13]). State of oral hygiene: Assessment by Clinical Oral Assessment Chart (Table 1). RSST: Repetitive Saliva Swallowing Test; ODK: Oral diadochokinesis; KCL: Kihon checklist. *: There is a significant relationship between social withdrawal and oral hypofunction. *p*-value: Chi-square test or Mann–Whitney U test (Age).

**Table 3 ijerph-17-08904-t003:** Relationship between the oral condition and physical factors.

	Oral Frail Condition							
	Robust	(%)	Pre-Frailty	(%)	Frailty	(%)	*p*-Value	
Participant	215	50.4	144	33.7	68	15.9		
Age *	71.5	5.3	72.8	5.3	75.8	6.8	*p* < 0.001	a, b
Gender Male	72	33.5	54	37.5	26	37.7	0.679	
Female	143	66.5	90	62.5%	43	62.3		
BMI	22.5	2.8	22.6	3.0	22.4	2.5	0.763	
SMI	6.5	0.9	6.5	1.0	6.3	0.7	0.203	
Body fat	27.1	7.3	27.0	6.9	27.6	7.7	0.827	
MMSE *	28.5	1.7	28.3	1.9	27.2	4.0	*p* < 0.001	a, b
Walking speed (m/sec) *	1.52	0.24	1.47	0.21	1.41	0.26	0.001	a
High knee extension (N) *	383.7	114.9	367.9	126.6	332.2	100.6	0.021	a

*: Significant difference in Kruskal–Wallis test. a: There is a significant difference between Robust and Frailty, b: There is a significant difference between Pre-frailty and Frailty. The following six items in Table 2 were used in assessing the degree of oral frailty: The number of remaining teeth, gingival condition, occlusal force, masticatory efficiency, and swallowing and dry mouth in KCL. Those who had reduced function in three or more items were assessed as frailty; those who had reduced function in 1–2 items were assessed as pre-frailty; and those who did not have reduced function in any item were assessed as robust. SMI: Skeletal muscle mass index, BMI: Body mass index, MMSE: Mini-Mental State Examination. *p*-value: Kruskal–Wallis test. Multiple comparison: Mann–Whitney U test; the *p*-value was adjusted by Bonferroni correction.

**Table 4 ijerph-17-08904-t004:** The results of social withdrawal score.

		Follow-Up (num)			All
	Social Withdrawal Score	Score 0	Score 1	Score 2	
Baseline	Score 0	286	**40**	**10**	336
(num)	Score 1	45	33	**5**	83
	Score 2	2	3	1	6
All		333	76	16	425

Data shows the number of people who answered the questions during the baseline and the follow-up survey. Negative answers to each of the social withdrawal questions were assigned a score of 1: “Do you go out at least once a week?” (Yes = 0/No = 1); and “Compared to the last year, do you go out less frequently?” (Yes = 1/No = 0). Participants who had a total score of 1 or greater were considered to have a social withdrawal tendency. Bold type participants indicate the subject with “worse” group for social withdrawal.

**Table 5 ijerph-17-08904-t005:** Factors related to social withdrawal.

Explanatory Variables	B	Standard Error	Wald	*p*-Value	Hazard Ratio	95% CI of the Hazard Ratio	
						Lower	Upper
Age	0.01	0.02	0.17	0.682	1.01	0.97	1.04
MMSE	0.01	0.04	0.05	0.820	1.01	0.94	1.08
walking speed *	−1.12	0.47	5.66	0.017	0.33	0.13	0.82
High knee extension *	-0.002	0.0009	3.71	0.054	1.00	1.00	1.00
Oral fraility *							
Robust	-	-	3.99	0.136	-	-	-
Pre-frailty	0.29	0.25	1.43	0.231	1.34	0.83	2.17
Frailty	0.60	0.30	3.90	0.048	1.82	1.00	3.29

A Cox proportional hazard model was used. Duration (the number of days) from baseline to the second examination was set as the time variable; the worsening of social withdrawal was set as the objective variable; confounding factors that were found to be significantly associated with oral frailty and the state of oral frailty were set as the explanatory variables. The relationship between social withdrawal and oral hypofunction was examined with the above conditions (force entry). *: Statistically significant explanatory variables.

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
