# Peer review of "Does Oral Hypofunction Promote Social Withdrawal in the Older Adults? A Longitudinal Survey of Elderly Subjects in Rural Japan"

_ijerph, 2020, doi:10.3390/ijerph17238904_

Round 1

Reviewer 1 Report

This paper addresses social withdrawal as a function of oral hypofunction.

In the Abstract, delete the word "exacerbates" and "exacerbation of" (lines 27, 34). Similarly, delete the words "exacerbates" and "exacerbation of" throughout the paper for the same reason. Exacerbates implies that social withdrawal has already occurred and is worsening; that has not been established in the paper.

line 28: change "remaining teeth" to "number of remaining teeth"

It is confusing to begin the Introduction section with a description of the implications of falling, without first having discussed what the term oral hypofunction implies and what its sequellae might be.  It would be helpful for the authors to define what they mean by using the term "oral hypofunction".  Is it tooth loss?  If so, there are additional references (cf. International Dental Journal) that support the impact of tooth loss on longevity/lifespan.

line 59: include "ability" as well as "motivation" to go out.  Halitosis is an additional factor impairing socialization (not mentioned in this section).

In Section 2.1, please add the percentage of adults that comprised the study sample.

A copy of the Kihon checklist (KCL) should be included as an Appendix, with appropriate citation, for those not familiar with it.

In Section 2.2, please describe how the three questions highlighted were used. 

Section 2.4, which discusses the evaluation of oral hypofunction, is somewhat confusing.  the paragraph beginning on line includes masticatory force, tongue pressure, and swallowing performance.  It would be helpful to have an introductory statement explaining what is being included in the paragraph, and what those values reflect.

Section 2.5, line 171: please change the word "worsening" to "status of".  Using the word "worsening" reflects a bias on the investigators' part.

Table 3: Title should be Oral condition status.  It seems to only include females?  Where is the male data, and where is the comparison?

Table 4, and earlier in the paper, it is confusing to assign a value of 1 for a positive response to the question "Do you go out at least once a week?" and then use a score of 1 or greater to be considered as having social withdrawal tendency.  This is a flawed inference.

Section 4.1 contents should be placed in the Introduction to help the reader understand the context of this study.

Section 4.2: The authors use the term "oral frailty" in this section.  The title of the manuscript includes the term "oral hypofunction". Are the authors using these terms synonymously?  It is confusing for the reader, and terminology should be consistent throughout the manuscript (title and contents).

Section 5: It would strengthen the manuscript to also discuss quality of life as a related concept. In addition, suggesting "maximizing" or "optimizing" oral function as a conclusion might be helpful.

Author Response

This paper addresses social withdrawal as a function of oral hypofunction.

In the Abstract, delete the word "exacerbates" and "exacerbation of" (lines 27, 34). Similarly, delete the words "exacerbates" and "exacerbation of" throughout the paper for the same reason. Exacerbates implies that social withdrawal has already occurred and is worsening; that has not been established in the paper.

Response:
We revised "exacerbate/exacerbation" to “worsening” throughout the text.

line 28: change "remaining teeth" to "number of remaining teeth"

Response:
We revised "remaining teeth" to “number of remaining teeth” throughout the text.

It is confusing to begin the Introduction section with a description of the implications of falling, without first having discussed what the term oral hypofunction implies and what its sequellae might be. It would be helpful for the authors to define what they mean by using the term "oral hypofunction". Is it tooth loss? If so, there are additional references (cf. International Dental Journal) that support the impact of tooth loss on longevity/lifespan.

Response:
Oral hypofunction is not equal to tooth loss; rather, tooth loss is a factor that causes oral hypofunction. As suggested, we clarified the definition of oral hypofunction and its sequelae in the INTRODUCTION (page 2 Line 60-61).

line 59: include "ability" as well as "motivation" to go out. Halitosis is an additional factor impairing socialization (not mentioned in this section).

Response: As suggested, we mentioned the relationship between halitosis and social withdrawal (page 2 Line 73).

In Section 2.1, please add the percentage of adults that comprised the study sample.

Response: As suggested, we added the percentage of adults that comprised the study sample (page 3 Line 97).

A copy of the Kihon checklist (KCL) should be included as an Appendix, with appropriate citation, for those not familiar with it.

Response: As suggested, we added the Kihon Check List as “Supplement file 1 “.

In Section 2.2, please describe how the three questions highlighted were used. 

Response:
The Kihon checklist is a self-administered evaluation created by the Ministry of Health, Labor and Welfare of Japan and is a screening tool introduced when the long-term care insurance system was revised in 2006. The Kihon checklist’s questionnaire consists of 25 questions in the following 7 areas: activities of daily living, social activities of daily living, physical function, nutritional status, oral function, social activities of daily living (social withdrawal), cognitive function, and depressed mood. In the present study, we selected two social withdrawal-related questions and three questions about the oral function related to oral frailty to create our evaluation items.

We have now added the above explanations to the 2.2 Survey questionnaire section (page 3 Line108-115).

Section 2.4, which discusses the evaluation of oral hypofunction, is somewhat confusing. the paragraph beginning on line includes masticatory force, tongue pressure, and swallowing performance. It would be helpful to have an introductory statement explaining what is being included in the paragraph, and what those values reflect.

Response: We have now added an introductory statement explaining what is included and what those values reflect in Section 2.4 (page 3 Line140-144).

Section 2.5, line 171: please change the word "worsening" to "status of". Using the word "worsening" reflects a bias on the investigators' part.

Response: As suggested, we changed the word "worsening" to "status of" (page 6 Line 198).

Table 3: Title should be Oral condition status. It seems to only include females? Where is the male data, and where is the comparison?

Response: As suggested, we changed Table 3’s title from “oral frailty” to “oral condition”. We also revised the table to include data from males and p-values.

Table 4, and earlier in the paper, it is confusing to assign a value of 1 for a positive response to the question “Do you go out at least once a week?” and then use a score of 1 or greater to be considered as having social withdrawal tendency. This is a flawed inference.

Response: There were some errors in our description of scoring. We have now revised section 2.2 (page 3 Line 114-116) and the Table 4 legend.

Section 4.1 contents should be placed in the Introduction to help the reader understand the context of this study.

Response: As suggested, we have now moved this text to the introduction (page 3 Line 52-58).

Section 4.2: The authors use the term “oral frailty” in this section. The title of the manuscript includes the term “oral hypofunction”. Are the authors using these terms synonymously? It is confusing for the reader, and terminology should be consistent throughout the manuscript (title and contents).

Response: As you note, we do not consider oral frailty and oral hypofunction to be synonyms. We have now explained the terms in the Introduction and revised them throughout the text with the aim of unifying our usage of the words (Page 2 Line 52-62).

Section 5: It would strengthen the manuscript to also discuss quality of life as a related concept. In addition, suggesting "maximizing" or "optimizing" oral function as a conclusion might be helpful.

Response: As suggested, we have now discussed the QOL according to the indication and mentioned "maximizing" or "optimizing" the oral function in the conclusion (Page 9 Line 319-321).

Reviewer 2 Report

Oral hypofunction troubled most of the elderly while falling-induced events really threated to the old people. Therefore, it is interesting that Hasegawa Y et al. studied the relationship between oral hypofunction and fall-related physical functions via social withdrawal. Totally, the study were well designed and the results were acceptable to support their conclusions. However, some points should be improved.

Minor points:

  1. In all the tables, “%” can be deleted from the data wells because the authors have shown in the most top well, which might make the tables more clean and neat.
  2. In table 2, there is no p value in the female group. And the authors should show the result when combine all the males and females.
  3. I think the author can discuss if the impaired physical ability of the elderly was partially resulted from the social withdrawal caused by oral hypofunction.

Author Response

Reviewer 2

Oral hypofunction troubled most of the elderly while falling-induced events really threated to the old people. Therefore, it is interesting that Hasegawa Y et al. studied the relationship between oral hypofunction and fall-related physical functions via social withdrawal. Totally, the study were well designed and the results were acceptable to support their conclusions. However, some points should be improved.

Response: Thank you for your helpful peer review. According to your comments, the text has been revised.

Minor points:

  1. In all the tables, “%” can be deleted from the data wells because the authors have shown in the most top well, which might make the tables more clean and neat.

Response: As suggested, we deleted the % values and arranged the tables to make them easier to understand.

  1. In table 2, there is no p value in the female group. And the authors should show the result when combine all the males and females.

Response: As suggested, we have now revised Table 2. Regarding gender, the p-value of the result of the chi-square test is shown (P=0.007).

  1. I think the author can discuss if the impaired physical ability of the elderly was partially resulted from the social withdrawal caused by oral hypofunction.

Response: As suggested, we have now discussed the causal relationship between the progression of social withdrawal and oral hypofunction in the discussion (Page 9 Line 291-294).

Reviewer 3 Report

This is a very interesting and well-written manuscript. There are a few minor issues the authors should consider.

Introduction: since the focus is social withdrawal and oral hypofunction it would be more appropriate if the introduction started with these subjects instead of falling.

Please insert an explanation of oral hypofunction and if possible how common it is. You have an explanation in the discussion section, which fits better in the introduction.

Material and methods: Are the subjects included community-dwelling or could also nursing home residents participate? Were persons interested in participation contacted by phone?

Results: line 131. Please clarify if only one item needed to be classified as cautious to get oral hypofunction.

Author Response

Reviewer 3

 This is a very interesting and well-written manuscript. There are a few minor issues the authors should consider.

Response: Thank you for your helpful peer review. As suggested, the text has been revised.

Introduction: since the focus is social withdrawal and oral hypofunction it would be more appropriate if the introduction started with these subjects instead of falling.

Response: The major cause of social withdrawal is falling, so we have mentioned this point in the Introduction. We added explanations of oral frailty and oral hypofunction to the introduction to make these easier to understand (Page 2 Line 52-53, 60-62).

Please insert an explanation of oral hypofunction and if possible how common it is. You have an explanation in the discussion section, which fits better in the introduction.

Response: As suggested, we have now added an explanation of oral hypofunction and moved the description of this from the Discussion to the Introduction (Page 2 Line 52-58).

Material and methods: Are the subjects included community-dwelling or could also nursing home residents participate? Were persons interested in participation contacted by phone?

Response: This study was only conducted among elderly who were independent. The first recruitment was as described, and for the second survey, we contacted only those who did not apply in two years after from baseline survey. We have now added the above description to the Methods section (Page 3 Line 99-100).

Results: line 131. Please clarify if only one item needed to be classified as cautious to get oral hypofunction.

Response: As suggested, we have now clarified this point (Page 5 Line 155).
